# Investigating the Antibacterial Activity and Safety of Zinc Oxide Nanoparticles versus a Commercial Alcohol-Based Hand-Sanitizer: Can Zinc Oxide Nanoparticles Be Useful for Hand Sanitation?

**DOI:** 10.3390/antibiotics11111606

**Published:** 2022-11-11

**Authors:** Aliaa Ismail, Nermeen R. Raya, Ahmed Orabi, Alaa M. Ali, Yasmin Abo-zeid

**Affiliations:** 1Department of Pharmaceutics and Industrial Pharmacy, Faculty of Pharmacy, Helwan University, Cairo 11795, Egypt; 2Helwan Nanotechnology Center, Helwan University, Cairo 11792, Egypt; 3Microbiology Department, Faculty of Veterinary Medicine, Cairo University, Giza 11221, Egypt; 4Department of Pathology, Faculty of Veterinary Medicine, Cairo University, Giza 11221, Egypt

**Keywords:** zinc oxide nanoparticles, hospital-acquired infections, Sterillium, hand sanitation

## Abstract

Hand hygiene is the key factor to control and prevent the spread of infections, for example, hospital-acquired infections (HAIs). People commonly use alcohol-based hand sanitizers to assure hand hygiene. However, frequent use of alcohol-based hand sanitizers in a pandemic situation (e.g., COVID-19) was associated with serious drawbacks such as skin toxicity including irritation, skin dermatitis, and skin dryness or cracking, along with peeling, redness, or itching with higher possibility of infection. This demands the development of alternative novel products that are effective as alcohol-based hand sanitizers but have no hazardous effects. Zinc oxide nanoparticles (ZnO-NPs) are known to have broad-spectrum antimicrobial activity, be compatible with the biological system and the environment, and have applicable and economic industrial-scale production. Thus, ZnO-NPs might be a good candidate for hand sanitation. To the best of our knowledge, the antibacterial activity of ZnO-NPs in comparison to alcohol-based hand sanitizers has not yet been studied. In the present work, a comparative study of the antibacterial activity of ZnO-NPs vs. Sterillium, a commercial alcohol-based hand sanitizer that is commonly used in Egyptian hospitals, was performed against common microorganisms known to cause HAIs in Egypt, including *Acinetobacter baumannii, Klebsiella pneumoniae, Methicillin-resistant Staphylococcus aureus* (MRSA), and *Staphylococcus aureus*. The safety profiles of ZnO-NPs and Sterillium were also assessed. The obtained results demonstrated the superior antibacterial activity and safety of ZnO-NPs compared to Sterillium. Therefore, ZnO-NPs could be a promising candidate for hand sanitation in comparison to alcohol-based hand sanitizers; however, several studies related to long-term toxicity and stability of ZnO-NPs and investigations into their antimicrobial activity and safety in healthcare settings are still required in the future to ascertain their antimicrobial activity and safety.

## 1. Introduction

Hospital- and community-acquired infections are escalating and pose a serious public health problem worldwide [1,2]. Hands are considered to be the primary route for transmitting microbes and infections to individuals [3,4,5]. This was further confirmed by the COVID-19 pandemic, which originated in December 2019 [6,7]. Therefore, the Centers for Disease Control and Prevention and the World Health Organization stated that hand hygiene is the key action to control/prevent the spread of infections in healthcare settings and the community [4]. Washing hands with soap and water is the traditional strategy to maintain hand hygiene; however, in healthcare settings, alcohol-based hand sanitizers are considered the most effective hand sanitizer products due to their broad-spectrum antimicrobial activity and ability able to kill micro-organisms including bacteria, parasites, fungi, and viruses [8,9]. Frequent application of alcohol-based hand sanitizers, especially in the occasion of a pandemic (e.g., the COVID-19 pandemic), was associated with several challenges and concerns such as skin toxicity including irritation, skin dermatitis, skin dryness or cracking, and peeling, redness, or itching [10], which was associated with an increased risk of infection [11]. In some cases, prolonged exposure to alcohol-based hand sanitizers was reported to be accompanied by an increased incidence of viral outbreaks, e.g., norovirus [11,12,13], and antimicrobial resistance [11,14,15]. This demands the development of alternative novel products that are as effective as alcohol-based hand sanitizers, but with no toxic effects.

Nanotechnology is a cutting-edge science that has been applied in different aspects for the development of human life, e.g., agriculture, health, electronics, textiles, and medicine. Nanoparticles are particles sized from 1 to 1000 nm and are used to improve the therapeutic activity of conventional medicines and overcome their side effects [16,17,18,19,20,21,22,23,24], as well as for diagnostic purposes [25]. Moreover, nanoparticles were reported to have antimicrobial activity against viruses [7,26,27,28,29,30] and bacteria [2,7,31].

Metal oxide nanoparticles were reported to have broad-spectrum antimicrobial activity [27,32]. Furthermore, they are effective against resistant bacteria and, thus, are an excellent model to control/prevent hospital-acquired infections (HAIs). Their main antimicrobial mechanism of action involves generating reactive oxygen species (ROSs) that are able to attack multiple sites (e.g., proteins, nucleic acids, lipids, enzymes, etc.), thus microorganisms are not able to develop resistance against metal-oxide nanoparticles [32,33].

The broad-spectrum antimicrobial activity of zinc oxide nanoparticles (ZnO-NPs) has been previously reported [32,34]. Moreover, ZnO-NPs are reported to be compatible and non-toxic to the biological system and the environment [35,36]. In addition, their production is very economic and can be easily applied at an industrial scale [35,37]. Overall, the application of ZnO-NPs has been promoted in different disciplines [35].

Due to the broad-spectrum antimicrobial activity and high safety of ZnO-NPs, we hypothesized that they will be an excellent alternative for alcohol-based hand sanitizers in healthcare settings and the community. However, to the best of the authors’ knowledge, the antibacterial activity and biosafety of ZnO-NPs in comparison to alcoholic-based hand sanitizers have not been studied yet. Therefore, the current study is considered the first assessment of the antibacterial activity and safety of ZnO-NPs compared to a commercial alcohol-based hand sanitizer (Sterillium) that is commonly used in Egyptian hospitals and communities [38]. In the current study, ZnO-NPs were synthesized chemically, and the antibacterial activity of their colloidal solution in comparison to Sterillium was investigated against multidrug-resistant bacteria that are commonly known to cause HAIs in Egypt: *Acinetobacter baumannii (A. baumannii), Klebsiella pneumoniae (K. pneumoniae),* Methicillin-resistant *Staphylococcus aureus* (MRSA), and *Staphylococcus aureus (S. aureus)* [39,40,41]. The cytotoxicity of these formulations to human dermal fibroblast cells was also assessed to ascertain the biosafety (in vitro) of ZnO-NPs to replace alcohol-based hand sanitizers in healthcare settings. This was followed by investigating their sanitation effect on the shaved skin of rats contaminated with *S. aureus*. The obtained results demonstrated the superior antibacterial activity and safety of ZnO-NPs compared to Sterillium. Therefore, ZnO-NPs are recommended as a replacement for alcohol-based hand sanitizers. However, a clinical study in healthcare settings still needs to be performed in the future to ascertain the efficacy and safety of ZnO-NPs as a hand sanitizer.

## 2. Materials and Methods

### 2.1. Materials

Zinc nitrate hexahydrate (ZnNO_3_·6H_2_O) with a 98% analytical grade was purchased from LOBACHEMIE PVT.LTD (Mumbai, India). A marketed hand sanitizer product (Sterillium^®^) manufactured by the HARTMANN GROUP’s manufacturing company (Germany) was purchased from a local store. All other reagents were supplied by Sigma-Aldrich (Germany). All chemicals and reagents were of analytical grade. Double-distilled water was obtained from the New Water Purification System (New Human Power I, VER. 2.0, Seoul, Korea)

Microbiological media and all chemical reagents were purchased from Oxoid, UK. Four hospital-acquired human pathogenic bacteria, namely, *Acinetobacter baumannii* (*A. baumannii*), *Klebsiella pneumoniae* (*K. pneumoniae*), *Methicillin-resistant Staphylococcus aureus* (MRSA), and *Staphylococcus aureus* (*S. aureus*), were collected from a clinical setting in Cairo, Egypt. They were identified as multidrug-resistant bacteria.

The adult human dermal fibroblast (HDFa) cell line (PCS-201-012, HDFa) was purchased from the American Type Culture Collection (ATCC). The EMEM medium, fetal bovine serum (FBS), non-essential amino acids (NEAAs), penicillin, and streptomycin were supplied by Thermo Fisher (Hessen, Germany). The MTT reagent was supplied by Thermo Fisher, Germany. All other materials were purchased from Sigma-Aldrich and used as supplied.

### 2.2. Methods

#### 2.2.1. Synthesis of ZnO-NPs

ZnO-NPs were synthesized by the chemical co-precipitation method following a previously published protocol [42] with the following modifications. Briefly, the zinc nitrate hexahydrate solution (20 mL, 0.1 M, double-distilled water) was prepared and mixed with a solution of PEG600 (20 mL, 0.0083 M, double-distilled water) for 5 min at room temperature. A sodium hydroxide solution (0.2 M) was added dropwise to the previous mixture while stirring until the required pH was reached. The mixture was stirred at the investigated temperature for 90 min. The white precipitate produced by the reaction was recovered by centrifugation (Bio Lion, model: HC-HR20, Manassas, VA, USA) at 4 °C and 12,000× *g* rpm for 5 min. The precipitate was washed three times with double-distilled water (30 mL) to dispose of unreacted salts and excess base. The dry powder of ZnO-NPs was obtained after drying the washed precipitate in an oven (Rumo model G25, Cairo, Egypt) at 150 °C for 6 h. The colloidal solution of ZnO-NPs for further experimental work was prepared by dispersing the powder in polyvinylpyrrolidone (PVP K25) (3% *w*/*v*, double-distilled water) and sonicating it for 10 min using a probe sonicator at an amplitude of 60 (Hielscher-Ultrasound Technology-UP50H, Teltow, Germany).

##### Experimental Design and Statistical Analysis by a Full Factorial Design

In order to obtain ZnO-NPs with the smallest particle size and a homogenous distribution, a two-level full factorial (2^3^) experimental design was employed using DESIGN EXPERT^®^ software version 12 (Stat-Ease Inc. Minneapolis, MN, USA) to study the effect of the (1) reaction pH (X1), (2) reaction temperature (X2), and (3) reaction stirring rate (X3) on the targeted responses for ZnO-NPs and their (1) particle size (Y1), (2) polydispersity index (Y2), and (3) surface charge (Y3). Table 1 lists the design factors and levels considered, which were determined by a literature survey and preliminary studies. Each sample was performed in triplicate, and the data for each response were presented as an average value ± standard deviation (SD). To assess the significance of the tested variables on the response, an ANOVA (analysis of variance) at *p* < 0.05 was conducted. The design was evaluated using polynomial equations in terms of coded factors to analyze and predict the main effect of each factor and the two-factor and three-factor interactions on the responses, as follows:Y = β_0_ + β_1_X_1_ + β_2_X_2_ + β_3_X_3_ + β_4_X_1_X_2_ + β_5_X_1_X_3_ + β_6_X_2_X_3_ + β_7_X_1_X_2_X_3_
where Y is the response variable, β_0_ is the intercept, and β_1_, β_2_, β_3_, β_4_, β_5_, β_6_, and β_7_ are the regression coefficients.

The terms X_1_, X_2_, and X_3_ represent the main effects of pH, reaction temperature, and stirring rate, respectively. X_1_X_2_, X_1_X_3_, and X_2_X_3_ are the two-factor interactions and X_1_X_2_X_3_ represents the three-factor interaction, both of which show how the response changes when two or three factors are concurrently changed. The best-fitting model for each response was selected based on the highest value of predicted R^2^ and the lowest value of predicted residual error sum of squares (PRESS) [43].

The 3-D response surface plots of the model-predicted responses were applied to specify the interactive relationships between the significant variables. DESIGN EXPERT^®^ software version 12 (Stat-Ease Inc. Minneapolis, MN, USA) was used to design the tests and perform statistical and graphical analyses of the obtained data.

#### 2.2.2. Characterization of ZnO-NPs

##### UV-Visible Spectral Analysis

A known concentration of ZnO-NPs was dispersed in 3% PVP K25 and scanned over a wavelength ranging from 300 to 1100 nm using a UV–Vis spectrophotometer (JASCO, V-630, Tokyo, Japan).

##### Particle Size and Zeta-Potential

The dried NPs were re-dispersed in the corresponding aqueous phase by probe sonication for 5 min (UP50H, Hielscher ultrasound technology, Teltow, Germany). The particle size and zeta potential were measured using Malvern Zeta sizer Nano ZS (Malvern Instruments Ltd., Malvern, UK). Samples were diluted with 3% PVP K25 to produce a count rate of 50 to 300 Kcps, and measurements were performed at 25 °C ± 0.1.

##### Transmission Electron Microscopy (TEM)

ZnO-NPs were imaged using TEM (H-700, Hitachi Ltd., Tokyo, Japan) at an accelerated voltage of 80 kv using a negative staining method following our previously published protocol [2,7]. Briefly, the sample was diluted (1:50) with double-distilled water and then a drop of the diluted sample was applied to a mesh copper grid coated with a carbon film and was left to dry for 5 min. After this, a drop of the phospho-tungstic acid solution (2% *w*/*w*) was added to the grid for 50 s and the excess liquid was removed using filter paper. The grid was left to air-dry in advance of imaging. TEM imaging was performed for the identification of the sample’s uniformity, shape, size, and presence of any aggregation.

##### Thermogravimetry and Differential Scanning Calorimeter (TGA/DSC)

Thermograms of ZnO-NPs were recorded on simultaneous DSC–TGA equipment (model SDTQ 600, London, USA) following the previously published protocol [44] with the following modifications: Powder of the sample (12 mg) was placed in aluminum pans and covered by aluminum lids. The thermal behavior of the sample was recorded at a scanning rate of 10 C/min under air conditions over a temperature range of 25–1000 °C. The instrument was calibrated using an indium standard.

##### Fourier Transformer Infra-Red (FTIR) Analysis

Infrared spectra of the synthesized ZnO-NPs powders were recorded using a PerkinElmer Frontier FT-IR spectrometer with the PerkinElmer Universal ATR sampling accessory (diamond window) in the wavenumber range of 4000 to 400 cm^−1^ and at a resolution of 4 cm^−1^, following a previously published protocol [45], to confirm their successful synthesis.

##### X-ray Diffraction (XRD) Analysis

The X-ray diffraction pattern of the synthesized ZnO-NPs (dry powder) was obtained by following a previously published protocol [46] in which the powdered sample was mounted onto the sample holder and analyzed by an X-ray diffractometer with a 2θ angle of 4–80°. Cu-Kα radiation (λ = 1.5406 Å) at a voltage of 40 kV and a current of 40 mA was used to confirm the presence of ZnO-NPs and the analysis of the structure.

#### 2.2.3. Antibacterial Study

##### Susceptibility Test

The susceptibility of *A. baumannii, K. pneumoniae,* MRSA, and *S. aureus* was evaluated by the agar diffusion method following the standard of the Clinical and Laboratory Institute [47].

##### Determination of Minimum Inhibitory Concentration (MIC)

The synthesized ZnO-NPs and Sterillium were screened for their antibacterial properties against a variety of Gram-positive (*Staphylococcus aureus* ATCC 12600 and MRSA ATCC 25923) and Gram-negative bacteria (*K. pneumoniae* ATCC 10031 and *A. baumanni* ATCC 15308) by the micro dilution standard method using Mueller–Hinton broth. An inoculum of approximately 1.5 × 10^8^ colony-forming units from each strain was applied to the wells of 96 microtiter plates, and the plates were incubated at 37 °C for 18–24 h. The well with the lowest concentration of tested samples showing no growth was defined as the minimum inhibitory concentration (MIC). All organisms used in this study were standard strains obtained from the American Type Culture Collection (ATCC). All the MIC experiments were performed in triplicate. No differences in the readings (triplicates) were observed. A negative control (3% PVP vehicle carrying ZnO-NPs) and a positive control (bacteria grown in growth media only) were carried out for each bacterium.

##### Time–Kill Curve Study of ZnO-NPs Solution

A time–kill curve study was performed to determine the effective time of antibacterial activity of the ZnO-NPs solution. Briefly, an overnight culture of MRSA and *A. baumannii* adjusted to 5 log cfu/mL was inoculated with the MIC of the freshly prepared ZnO-NPs solution vs. Sterillium. The inoculum was taken at time intervals and serially diluted in dilution fluid (0.9 NaCl and 1% peptone). One hundred microliters of each dilution were then spread over the surface of well-dried TSA, followed by incubation for 18–24 h at 37 °C. Colonies were then counted, and the percentage of growth inhibition in comparison to controls was calculated. Two independent experiments were performed in triplicate in each experiment for each tested sample.

##### Transmission Electron Microscopy Study

Fresh bacterial growth cultures of MRSA and *A. baumannii* were separately grown overnight in the presence of ZnO-NPs and Sterillium at a concentration less than the MIC. Bacterial cells were washed twice with PBS (1.5 mL, pH 7.2) and then fixed with glutaraldehyde in PBS (2% *v/v*). The samples were post-fixed with OsO4 (1% *w*/*v*) in PBS (5 mmol/L) for 1 h at room temperature, washed three times with the PBS buffer, dehydrated in graded ethanol, and then embedded in epoxy resin.

Microtome sections were prepared for the specimens at approximately 500–1000 µm thickness using a Leica ultra-cut microtome (UCT microtome). These sections were then stained with toluidine blue and examined with a lens at a magnification power of 1X, and the sections were examined by a camera (Leica ICC50 HD). Ultrathin sections approximately 75–90 µm in thickness were then prepared and double-stained with saturated uranyl acetate and lead citrate. At the chosen magnification, the grids were examined with a JEOL Transmission Electron Microscope (JEM-1400 TEM, JEOL-Hitachi, Tokyo, Japan). Images were captured by an AMT CCD Optronics camera with 1632 × 1632-pixel format as a side-mount configuration. This camera used a 1394 FireWire board for acquisition.

#### 2.2.4. Cytotoxicity Assay

##### Cell Culture

HDFa cells were cultured in a complete EMEM culture medium containing FBS (10% *v/v*), Earle’s Balanced Salt Solution, non-essential amino acids, L-glutamine (2 mM), sodium pyruvate (1 mM), sodium bicarbonate (1500 mg/L), penicillin G sodium (10.000 UI), streptomycin (10 mg), and amphotericin B (25 µg), followed by the incubation of cells at 37 °C and 5% CO_2_, and the culture medium was refreshed every 24 h. As the density of cells was 70 to 90%, they were sub-cultured to achieve the desired density for the cytotoxicity test.

For the sub-culture, the cell culture medium was first removed and the flask was washed twice with PBS. To detach the cells from the culture flask, trypsin/EDTA was deposited; subsequently, this cell suspension was mixed with a fresh complete culture medium in Falcon tubes. Finally, the cells were collected via centrifugation at 1500× *g* rpm and then adjusted to the destiny required for the cytotoxicity test.

The MTT Cytotoxicity Assay was performed following our previously published protocol [2]; briefly, HDFa cells (4.5 × 10^3^ cells/well) were seeded into 96-well culture plates. The complete culture medium (100 µL; EMEM with FBS (10% *v/v*), Earle’s Balanced Salt Solution, nonessential amino acids, L-glutamine (2 mM), sodium pyruvate (1 mM), sodium bicarbonate (1500 mg/L), penicillin G sodium (10.000 UI), streptomycin (10 mg), and amphotericin B (25 µg) were added to the cells, followed by incubation for 24 h at 37 °C and 5% CO_2_. Tested samples were diluted with the complete culture medium to prepare a set of serial concentrations. Subsequently, the complete culture medium was removed from each well, and the diluted tested samples (100 µL) were added, followed by incubation for 24 h at 37 °C and 5% CO_2_. Then, the culture medium containing the formulations was removed, and a fresh complete culture medium (100 µL) was added. The MTT reagent (20 µL, 1 mg/mL) was added to each well, followed by incubation for 4 h at 37 °C and 5% CO_2_. This was followed by carefully removing the complete culture medium and adding the solubilizing agent to dissolve formazan crystals followed by calculating the percentage of cell viability. A set of control samples was prepared, and untreated HDFa cells and cells treated with blank samples (3% PVP, vehicle used to disperse ZnO-NPs) were also tested under the same conditions.

#### 2.2.5. In Vivo Study

##### Animals

Wistar rats (CLAVCAP-VACSER, Cairo, Egypt), weighing 180–200 gm were utilized in the current experiment to investigate the sanitation effect of ZnO-NPs vs. the commercial product (Sterillium). Rats were maintained under managed conditions with a 12 light–dark cycle at 25 ± 2 °C and 50% ± 20% relative humidity. They had free access to food, a standard commercial pellet diet (containing, at the very least, 5% fiber, 20% protein, 3.5% fat, 6.5% vitamins, and ash mixture), and were offered water ad libitum. For the whole experimental period, rats were kept in good hygienic conditions, and they were left for one week to acclimatize to the lab conditions before the onset of the experiment. All the procedures in this research were approved by the Institutional Animal Care and Use Committee, Faculty of Veterinary Medicine, Cairo University (Vet CU 20092022457). All efforts were made to minimize animals’ suffering.

##### Experimental Design

The skin hair (2.5 × 2.5 cm^2^) of twenty-five rats was removed one day prior to the experiment, and rats were randomly allocated into 5 groups with 5 rats in each: Group I: Negative control, untreated group, i.e., shaved skin was not swapped with either bacterial solution or any tested sample.;Group II: Positive control; shaved skin of rats was swapped with *S. aureus* (0.5 mL/rat, 1.5 × 10log8), without swapping any solution afterwards; Groups III, IV, and V: Shaved skin of rats was swapped by *S. aureus* (0.5 mL/rat, 1.5 × 10log8), followed by swapping with 1 mL of ZnO-NPs (180 µg/mL) colloidal dispersion, 3% PVP, the vehicle used to disperse ZnO-NPs, and Sterillium for Groups III, IV, and V, respectively, and this was repeated daily for 5 days. At the end of the experiment, the skin area was dissected out, wiped of blood, and kept in 10% neutral buffer formalin for histopathological investigations.

##### Histopathological and Immunohistochemical Evaluations

In a routine manner, formalin-fixed skin specimens were dehydrated in different grades of alcohol followed by clearance in xylol and embedding in paraffin. Serial sections 4–5 μm thick were obtained from the prepared paraffin blocks followed by their staining with Hematoxylin and Eosin (H&E) [48]. Histopathological findings in the skin were graded semi-quantitatively according to [49] with some modifications (0 = no abnormality, 1 = slight, 2 = mild, 3 = moderate) for each of the seven findings—hypertrophy, hyperkeratosis, parakeratosis, erosion, inflammatory cell infiltration, extracellular edema, and ulcers.

##### Biochemical Analysis

Reduced glutathione content (GSH) [50] and superoxide dismutase (SOD) [51] were determined.

##### Hematological Examination

A complete blood picture (CBC) and blood film examination were performed for the detection of any abnormality in the shape of RBCs and the incidence of cell hemolysis (if any).

#### 2.2.6. Statistical Analysis

All statistical analyses were performed using the Analysis of Variance (ANOVA). Analyses were carried out using GraphPad Prism 9.0 software at a confidence level of 95%. For nonparametric data analysis, the Kruskal–Wallis H test was performed, followed by the Mann–Whitney U test.

## 3. Results and Discussion

### 3.1. Characterization of ZnO-NPs

UV–Visible spectrum of the ZnO-NPs showed an absorption peak at 368 nm (Appendix A) and this matched what was reported in the literature, as ZnO-NPs had a strong absorbance at a wavelength of 310–385 nm [52,53]. This confirmed the successful synthesis of ZnO-NPs.

ZnO-NPs’ thermal stability is presented in Appendix A, in which ZnO-NPs show high thermal stability in terms of the overall weight loss obtained at the end of the testing time and temperatures up to 1000 °C. In particular, ZnO-NPs showed only 0.2187 wt% weight loss at temperatures up to 200 °C, which might be explained by the loss of water molecules that are bound to the surface of ZnO-NPs and is consistent with the literature [54]. However, the total weight loss at the end of the test was 2.459 wt%, which might be due to the release of water molecules that are chemically combined with PEG in addition to the decomposition of PEG 600 molecules adsorbed on the surface of ZnO-NPs. This matched what was reported in the literature [55].

FTIR spectra of ZnO-NPs are presented in Appendix A, and the bands at approximately 500 and 435.8 cm^−1^ are attributed to the formation of the stretching vibration of metal–oxygen (Zn–O) bonds. This matched the literature in which the infrared characteristic bands of ZnO-NPs were reported to be in the region of 680 up to 300 cm^−1^ [56]. The band recorded at 3421.7 cm^−1^ was characteristic of the stretching vibration of the intermolecular hydrogen bond (OH) existing between the adsorbed water molecules [45]. In addition, this band could be attributed to PEG molecules adsorbed onto the surface of ZnO-NPs [57,58], while the bands at 1617.8 and 1033 cm^−1^ could be correlated to the stretching vibration of (-C-O-C-) ether groups and (COO-) carboxylate groups, respectively, for PEG molecules [58,59].

An X-ray diffractogram of ZnO-NPs is presented in Appendix A, in which the diffraction peaks are located at 31.70°, 34.34°, 36.17°, 47.44°, 56.53°, 62.77°, 66.27°, 67.88°, 69.00°, 72.48°, and 76.90°. These peaks are in agreement with those reported in the literature [60] indicating the successful synthesis of ZnO-NPs in a crystalline form free of impurities.

#### 3.1.1. Full Factorial Design Analysis

A full factorial design (2^3^) was used to identify the optimum conditions to obtain ZnO-NPs with the smallest particle size, homogenous distribution, and highest stability as indicated by the zeta potential value. Experimental data values for the particle size, polydispersity index, and zeta potential obtained by the 2^3^ full factorial design are presented in Table 2. As revealed by Table 2, F3 was the only formula that met the required criteria, and its particle size, PDI, and zeta potential values were 496.27 ± 20.11 nm, 0.23 ± 0.04, and −16.20 ± 0.70, respectively. It is worth noting that different concentrations of PVP25K were investigated and only 3% PVP gave the lowest particle size and the highest stability (Appendix A).

Three variables were selected to optimize the size, polydispersity index, and zeta potential of synthesized ZnO-NPs, namely, pH (X_1_), reaction temperature (X_2_), and stirring rate (X_3_), and the interaction effect of the chosen parameters on the response observed in the experimental runs will be discussed below.

#### 3.1.2. Effect of Synthesis Process Variables on Particle Size of ZnO-NPs (Y_1_)

As presented in Table 2, the mean particle size for the synthesized ZnO-NPs ranged from 379.60 ± 39.80 to 775.80 ± 24.70 nm. The two-factors interaction (2FI) model was selected as the best-fitting model for particle size as it had the highest values of predicted R^2^ = 0.9490. Statistical analysis using ANOVA revealed that the model terms (X_1_, X_2_, X_3_, X_1_X_2_, X_1_X_3_, and X_2_X_3_) had a significant impact on particle size with *p* < 0.05 and an F-value of 107.95. The equation relating the effect of formulation variables to the particle size in terms of coded values was:Particle size = +562.09 + 46.40 X_1_ − 71.79 X_2_ − 40.69 X_3_ − 26.86 X_1_X_2_ + 62.57 X_1_X_3_ + 84.15X_2_X_3_

Accordingly, from the regression coefficients, the individual effect of pH (X_1_) on the particle size was noticeably positive. Thus, increasing the pH from 8 to 10 led to enhanced growth and increased the particle size of ZnO-NPs. At pH ≥ 8, a pure form of ZnO-NPs was reported to be formed [61], which was mainly due to the high reducing power at an alkaline pH that resulted in the formation of ZnO-NPs with high purity and no impurities [62,63]. Thus, we selected two levels of pH to lie within alkaline pH, and it was observed that increasing the pH value is associated with an increase in the particle size. This might be attributed to the decrease in the time required for the deposition of ZnO-NPs crystals, promoting particle aggregation [64].

Regarding the effect of the reaction temperature (X_2_) and stirring rate (X_3_) on particle size, both had a negative effect on particle size; in other words, an increase in the reaction temperature and stirring rate resulted in nanoparticles with smaller particle size. The former might be attributed to the rapid consumption of Zn^+2^ ions [65,66] while the latter might be due to the increased transfer of energy to the reaction medium, thus causing it to spread out into smaller droplets [67].

Moreover, the interaction between pH and the stirring rate (X_1_X_3_) and the interaction between the reaction temperature and stirring rate (X_2_X_3_) had positive effects on the particle size of ZnO-NPs. On the other hand, the interaction between pH and the reaction temperature (X_1_X_2_) had a negative effect on particle size. The 3-D response surface plots for (X_1_X_2_), (X_1_X_3_), and (X_2_X_3_) on the particle size of ZnO-NPs are illustrated in Figure 1.

#### 3.1.3. Effect of Synthesis Process Variables on Homogeneity of ZnO-NPs (Y_2_)

The homogeneity (uniformity of particle size) can be assessed by the value of the polydispersity index (PDI). PDI ranges from 0 to 1 and is a measure of the broadness of the particle size distribution. PDI values of less than 0.7 give a general indication of the good quality of the particle size distribution of nanoparticles [68]. The obtained results for PDI values of ZnO-NPs ranged from 0.23 ± 0.04 to 0.50 ± 0.01 (Table 2).

The three-factor interaction (3FI) model was selected as the best-fitting model for PDI as it had the highest values of predicted R^2^ = 0.8403. Statistical analysis using ANOVA revealed that the model terms (X_1_, X_2_, X_3_, X_1_X_2_, X_1_X_3_, X_2_X_3_, and X_1_X_2_X_3_) had a significant impact on PDI with *p* < 0.05 and an F-value of 29.93. The equation relating the effect of formulation variables on the PDI in terms of coded values was:PDI = +0.3425 + 0.0437X_1_ − 0.0315X_2_ + 0.0266X_3_ − 0.0339X_1_X_2_ − 0.0232X_1_X_3_ + 0.0363X_2_X_3_ + 0.0178X_1_X_2_X_3_

Accordingly, from the regression coefficients, the individual effects of pH (X_1_) and stirring rate (X_3_) on PDI were positive. An increase in pH (X_1_) increased the settling rate of ZnO-NPs, and this involved less-controlled growth of particles and, hence, a higher value of PDI was obtained due to the formation of particles of different sizes [69]. An increase in the stirring rate from 900 to 1500 rpm led to increased PDI, which indicated the formation of a broad range of particle distributions. This might be explained by the occurrence of a non-uniform environment for the growth and nucleation of nanoparticles upon intense stirring (>1000 rpm) [70]. The two-factor interaction between the reaction temperature and stirring rate (X_2_X_3_) had a positive effect on PDI, while the interaction between pH and the reaction temperature (X_1_X_2_) and the interaction between pH and the stirring rate (X_1_X_3_) had a negative effect on PDI as shown in Figure 2. Moreover, the three-factor interaction between pH, reaction temperature, and stirring rate (X_1_X_2_X_3_) had a positive effect on PDI.

#### 3.1.4. Effect of Synthesis Process Variables on Zeta Potential of ZnO-NPs (Y_3_)

The values of zeta potential for the synthesized ZnO-NPs ranged from −9.94 ± 0.32 to −18.87 ± 0.15 mV (Table 2). The negative value could be attributed to the distribution of the pyrrolidone group of PVP K25 on the surface of ZnO-NPs, which provides a steric repulsive force between the nanoparticles [71].

The three-factor interaction (3FI) model was selected as the best-fitting model for zeta potential as it had the highest values of predicted R^2^ = 0.9215. Statistical analysis using ANOVA revealed that the model terms (X_1_, X_3_, X_1_X_2_, X_1_X_3_, X_2_X_3_, and X_1_X_2_X_3_) had a significant impact on zeta potential at *p* < 0.05 and an F-value of 63.26 while X_2_ was insignificant with *p* > 0.05 and an F-value of 0.1333. The equation related to the effect of formulation variables on the absolute values of zeta potential in terms of coded values was:Zeta potential = +14.09+ 0.8183X_1_ + 0.0517X_2_ − 0.3017X_3_ − 0.5575X_1_X_2_ + 1.41X_1_X_3_ + 1.37X_2_X_3_ + 1.98X_1_X_2_X_3_

Accordingly, from the estimated coefficients, the individual effect of the pH (X_1_) and reaction temperature (X_2_) on the zeta potential was positive. Regarding pH (X_1_), as the pH increased from 8 to 10, the absolute value of the zeta potential significantly increased (*p* < 0.05) [72]. A pH of 6.4 was previously reported to correspond to the isoelectric point of ZnO-NPs [72]. Thus, increasing pH beyond the value of the isoelectric point led to an increase in the surface charge of nanoparticles. Although the individual effect of the reaction temperature (X_2_) on the zeta potential was insignificant (*p* > 0.05), the positive coefficient of X_2_ indicated the effect of increasing the reaction temperature on increasing the absolute value of the zeta potential. This might be due to the modulation of the ionic distribution around the particles and thus, leading to alteration in potential distribution and increased zeta potential [73]. Moreover, the significant (*p* < 0.05) negative effect of the stirring rate (X_3_) on the zeta potential might be due to the agglomeration of individual particles that occurred at a high stirring rate leading to a reduction in the surface area, and hence, a lower surface charge was recorded [70,74].

The interaction between the reaction temperature and stirring rate (X_2_X_3_) and the interaction between pH and the stirring rate (X_1_X_3_) had a positive effect on the zeta potential, while the interaction between pH and the reaction temperature (X_1_X_2_) had a negative effect on the zeta potential as shown in Figure 3. Moreover, the interaction between the pH, reaction temperature, and stirring rate (X_1_X_2_X_3_) had a positive effect on the zeta potential.

#### 3.1.5. Optimization of the Synthesized ZnO-NPs

Optimization was carried out using a numerical optimization technique based on the desirability approach. According to the constraints provided in Table 1, Design Expert^®^ software revealed that the F3 formula had the optimum combination of synthesis parameter levels (pH of 8, reaction temperature of 70 °C, and stirring rate of 900 rpm). The F3 formula satisfied the required constraints with the highest desirability value, equivalent to 0.758. Furthermore, F3 was validated by comparing its actual experimental response values with its predicted response values. The percentage error for each response was determined and was found to be between 0 and 1.31% (Table 3).

#### 3.1.6. Transmission Electron Microscopy (TEM)

TEM was performed for the selected ZnO-NPs formulations, with F3 prepared at a concentration of 1 mg/1 mL. As presented in Figure 4, the TEM image shows spherical nanoparticles with no sign of particle aggregation. The particle size was smaller than that obtained by the Malvern Zeta-sizer instrument and ranged from 218 to 268 nm. The difference in particle size could be attributed to the different techniques applied to identify the particle size of ZnO-NPs.

### 3.2. Antibacterial Study

A sensitivity test was performed for four types of bacteria, and all proved to be resistant to some of the selected antibiotics, as revealed in the Appendix A. The MIC was determined to assess the antibacterial activity of the colloidal dispersion of ZnO-NPs and Sterillium against *S. aureus,* MRSA*, K. pneumonia,* and *A. baumannii*. Gram-positive bacteria were more susceptible to ZnO-NPs than Gram-negative bacteria as revealed by the MIC values. The MIC values recorded for *S. aureus* and MRSA were 180 and 350 µg/mL, respectively, which were lower than those recorded for *K. pneumoniae* and *A. baumannii* at 455 and 625 µg/mL, respectively. On the other hand, Sterillium recorded MIC values of 85, 68, and 59.5 (*v/v*%) for *S. aureus, K. pneumoniae,* and *A. baumannii*, respectively, while it was ineffective against MRSA.

MRSA and *A. baumannii* are the most common pathogens causing HAIs in Egypt [41,75]. Therefore, time–kill curves and TEM images for treated bacteria were performed only for those bacteria. The time–kill curves for ZnO-NPs and Sterillium against MRSA and *A. baumannii* are presented in Figure 5. As presented, both bacteria treated with a PVP solution (3% *w*/*v*, vehicle used to disperse ZnO-NPs) demonstrated an increase in bacterial count from 5 Log cfu/mL up to 8 Log cfu/mL, in contrast to the complete eradication of MRSA and *A. baumannii* 24 h after treatment with the ZnO-NPs colloidal dispersion. Therefore, the reduction in the bacterial count was solely attributed to ZnO-NPs. Sterillium was less effective than ZnO-NPs as it did not fully inhibit the growth of the bacteria, as the bacterial count recorded at 24 h for MRSA and *A. baumannii* was reduced from 5 Log cfu/mL to 4.4 ± 0.65 and 3.4 ± 0.53 Log cfu/mL, respectively.

The obtained results are consistent with a previous study [76,77] where the antibacterial activity of ZnO-NPs was tested against *S. aureus* and *E. coli* and showed MIC values of 1 and 3.4 mM, respectively. The reduced effect on Gram-negative bacteria was rationalized as the low permeability of lipopolysaccharide’s outer membrane that confers extra protection to the cell against external toxic compounds [2,78,79]. Moreover, Pati and colleagues [80] reported the higher susceptibility of *S. aureus* to ZnO-NPs than the Gram-negative bacteria *Mycobacterium bovis*-BCG, and again, this was also attributed to the presence of the extra-protective lipopolysaccharide outer membrane in Gram-negative bacteria. In contrast to Gram-negative bacteria, the peptidoglycan layer surrounding Gram-positive bacteria was reported to facilitate the entry of ZnO-NPs into bacterial cells, and this was followed by bacterial damage [81].

A TEM study was performed to identify the structural changes and suggest the possible antibacterial mechanisms of action for ZnO-NPs in comparison to Sterillium. TEM images for bacteria treated with ZnO-NPs and Sterillium vs. untreated bacteria are presented in Figure 6. Untreated MRSA and *A. baumannii* showed a normal cell shape, the cell wall appeared intact without any itches or degradation, and the cytoplasm and other intra-cellular components appeared normal without any abnormal structural changes identified. Upon treating MRSA with ZnO-NPs, rupturing of the cell walls was observed, leading to the release of cell cytoplasmic content. In contrast to MRSA treated with Sterillium, the bacterial cell appeared normal with no structural changes, and this is indicative of the inefficacy of Sterillium and its failure to eradicate MRSA. On the other hand, regarding *A*. *baumannii* treated with ZnO-NPs, a disruption to the cell wall was observed, and this might be responsible for the loss of intracellular contents, leading to bacterial death. Regarding *A. baumannii* treated with Sterillium, morphological changes were observed compared to untreated bacteria, and this involved the cell wall and the appearance of body projections, as indicated by black arrows in Figure 6.

TEM results were in agreement with previous studies performed to study the effect of ZnO-NPs on bacterial cell morphology on *S. aureus* [82], and *A. baumannii* [83], where the cell walls of both bacteria were ruptured. The latter observation was explained by the ability of ZnO-NPs to produce reactive oxygen species (ROSs). ROSs were argued to be responsible for the destruction of proteins and lipids of the bacterial cell wall followed by the release of intra-cellular components and, consequently, the death of bacteria [84,85]. Concerning Sterillium, based on its alcoholic constituents (2-Propanol, 1-Propanol, and Mecetronium etilsulfate), its antibacterial mechanism of action was due to the solubilization of the phospholipid bilayer of the bacteria, thereby increasing their permeability and the outflow of metabolites and intra-cellular components. This is reported to be associated with a change in the conformation of proteins embedded in the cell wall of bacteria, thus affecting transport and energy generation, ultimately leading to bacterial cell death [86]. However, it is worth noting that the frequent administration of alcohol-based hand sanitizers has been reported to be accompanied by many adverse effects on the skin, as previously discussed [10,87].

### 3.3. Cell Viability

The use of hand sanitizers has become a major tool, especially in clinical practice for the prevention of microorganisms commonly causing HAIs between healthcare workers and patients. This means that the hands of healthcare workers are required to be sanitized frequently each day, and this involves prolonged exposure to the chemicals in the hand sanitizer; thus, the biosafety of hand sanitizers is an essential issue to be addressed [3,88].

In the current study, we assessed the safety of ZnO-NPs and Sterillium in vitro using human dermal fibroblast cells. As presented in Figure 7A,B, the percentage of cell viability was concentration-dependent, where an increase in the sample concentration was accompanied by a reduction in the cell viability percentage.

The concentration responsible for the death of 50% of cells (CC50) was determined to assess the in vitro safety of ZnO-NPs and Sterillium. CC50 values for ZnO-NPs and Sterillium were 35.32 ± 1.86 µg/mL and 3.686 ± 0.195 *v/v*%, respectively.

To assess the possibility of using ZnO-NPs for hand sanitation, its MIC and CC_50_ values were compared to those recorded for Sterillium, a market product that is commonly used in Egyptian hospitals for hand sanitation as previously discussed. As revealed from Table 4, the MIC values of ZnO-NPs recorded for *S. aureus,* MRSA, *K. pneumoniae*, and *A. baumannii* were 5, 10, 13, and 17.7 times its CC50, respectively. However, it was reported that ZnO-NPs of up to 1000 mg were still considered compatible with the skin [89]. Thus, the highest concentration required from ZnO-NPs to assure complete eradication of bacteria is still considered safe to the skin with no harmful effects.

On the other hand, the MIC values of Sterillium against *S. aureus*, *K. pneumoniae,* and *A. baumannii* were 23, 18.5, and 16 times its CC50, respectively, while Sterillium was completely ineffective against MRSA. Sterillium is an alcohol-based hand sanitizer and, as discussed previously, the effective concentration of Sterillium required to eradicate bacteria is much higher than its CC50. This could explain the harmful effects on the skin reported with the frequent administration of Sterillium, such as skin irritation, redness, cracking [10], and the incidence of microbial resistance [11,14,15]. Thus, ZnO-NPs are more biocompatible and effective as an antibacterial agent than Sterillium. Therefore, ZnO-NPs can be expected to be applied safely for hand sanitation.

### 3.4. In Vivo Study

The gross findings for the five groups by the end of the experiment are presented in Figure 8. Group I (negative control, shaved animal skin was not swapped with *S. aureus*, without applying any tested solution) showed normal skin (Figure 8A). Groups II (positive control, animal skin was swapped only with *S. aureus*, and remained untreated) and IV (shaved skin was swapped with *S. aureus* and then swapped with 3% PVP, vehicle used to disperse ZnO-NPs) both showed dry, scaly, and thicker lesions that tend to darken in color, being brownish to blackish, and lichenification and fissures were observed (Figure 8B,D, respectively). These observations were commonly identified by the fourth or fifth day of the experiment due to bacterial infection on the skin. In advance of these scaly lesions, the skin showed erythema, swelling, and redness on the first and second days of the experiment. In contrast to Group III (shaved animal skin was swapped with *S. aureus* followed by swapping with ZnO-NPs), the skin appeared normal without any pathological alteration (Figure 8C) similar to the negative control group (Figure 8A) while Group V (shaved animal skin was swapped with *S. aureus* and then swapped with Sterillium), the skin showed mild erythema (Figure 8E).

#### 3.4.1. Histopathological Findings

The histopathological findings were identified and scored and are presented in Figure 9 and Table 5. The skin of Group I (negative, untreated group) appeared histopathologically normal (Figure 9A). In contrast, Group II (shaved skin was swapped with bacterial solution*, S. aureus*) showed severe histopathological alterations in all skin layers (Figure 9B), where both the dermis and epidermis thickened and were disrupted by severe necrosis on underlying tissues. Moreover, massive infiltration of pleomorphic inflammatory cells and macrophages was identified. In addition, hypogranulosis, spongiosis, and ulcers covered with necrotic tissue (crust) were found. Concerning Group III (swapped with ZnO-NPs dispersed in 3% PVP), the skin appeared completely normal without any histopathological alterations in either the epidermis or dermis (Figure 9C). Group IV (swapped with the vehicle used to disperse ZnO-NPs, PVP solution, 3% *w*/*v*), the skin reaction and histopathological examination were similar to that recorded in the positive control as presented in Figure 9D, and this is indicative that the vehicle used for ZnO-NPs did not have any antibacterial activity. Therefore, the antibacterial activity identified in Group III was solely correlated to ZnO-NPs. Regarding Group V (swapped with Sterillium), the histopathological investigations revealed infiltration of fewer macrophages and neutrophils (Figure 9E), indicating a mild inflammatory reaction.

#### 3.4.2. Hematological Examination

The mean values of the erythrogram are presented in Table 6. The statistical analysis of data revealed no significant differences (*p* > 0.05) between all groups. Additionally, the blood film presented in Figure 10 for all groups revealed no abnormalities or alterations to the shape of RBCs.

#### 3.4.3. Biochemical Analysis

The intracellular reduced Glutathione (GSH) and superoxide dismutase (SOD) were measured to identify the oxidative damage (if any) in all animal groups. The obtained results are presented in Figure 11, where Group II (positive control, animal skin was swapped only with *S. aureus*, and remained untreated) and Group IV (shaved skin was swapped with *S. aureus* and then swapped with 3% PVP) showed significantly (*p* < 0.05) lower values of dermal GSH content and SOD activity compared to other groups. On the other hand, Group III (shaved animal skin was swapped with *S. aureus* followed by swapping with ZnO-NPs) showed significantly (*p* < 0.05) higher values of both GSH and SOD compared to Group V (shaved animal skin was swapped with *S. aureus* followed by Sterillium swapping) and Group II. However, Group III showed a significantly (*p* < 0.05) lower value compared to Group I (negative control, shaved animal skin was not swapped with *S. aureus*, without applying any tested sample).

In humans, *staphylococcal* skin infections are fairly prevalent as these infections are reported to be resolved on their own [90]. However, in some circumstances, *S. aureus* can spread locally and develop serious, and occasionally deadly, deep infections [91,92]. This is due to the ability of *S. aureus* to penetrate the epidermal surface and disseminate through all skin layers when swapped on the skin [93]. Thus, in the current study, shaved rat skin was swapped with *S. aureus* and the bacteria was able to penetrate the skin barrier and produce deep infections, confirming the infectivity of the bacteria as revealed in gross examination (Figure 8). This was further confirmed by a histopathological study (Figure 9) where neutrophils, lymphocytes, histiocytes, and fibrin deposition were identified in both the positive control group (Group II) and the group treated with 3% PVP (Group IV). The histopathological study showed that inflammatory reactions affected the majority of the skin layers and, thus, our findings were comparable to the literature [94]. Moreover, crust formation was seen in Groups II and IV, and this also is in agreement with the literature [95]. This could be attributed to the ability of *S. aureus* to penetrate the skin barrier, and this triggered the inflammatory reactions as described previously. Furthermore, *S. aureus* is able to produce D-toxin that is able to trigger mast cell degranulation, raise Ig E levels, and enhance skin inflammation [96]. Moreover, *S. aureus* was reported to cause keratinocytes and generate cytokines, which cause skin injury and inflammation similar to acute dermatitis [97].

It is known that inflammatory cells are activated as a result of inflammation in animals, and this involves recruiting neutrophils and macrophages with reactive oxidants, such as hydrogen peroxide (H_2_O_2_), and oxygen radicals. These reactive oxygen substances produced by cells of the immune system show potent cytotoxic effects on pathogenic organisms [98]. Meanwhile, animals develop antioxidative systems to minimize oxidative damage by activating certain enzymes and releasing antioxidant factors. GSH and SOD are examples of this antioxidative system, as they are responsible for the detoxification of ROSs [32,99]. In our study, when rat skin was swapped with *S. aureus*, this was followed by triggering inflammatory cells as previously described, and consequently, antioxidative mechanisms were activated. This resulted in reducing the activity of SOD compared to the healthy rats in order to defend against the oxidative stress on the animal. Our findings were similar to those previously reported in the literature [98,100].

## 4. Conclusions

Hand hygiene is considered the key factor in controlling and preventing infection, either in hospital care settings or in the community. Alcohol-based hand sanitizers are commonly used by healthcare workers to assure hand hygiene to combat hospital-acquired infections, especially in pandemic situations (e.g., COVID-19). However, their frequent administration, especially during the COVID-19 pandemic, was associated with serious hazards such as skin toxicity, including irritation, skin dermatitis, skin dryness or cracking, and peeling, redness, or itching, creating a higher possibility of infection. Moreover, on some occasions, they were responsible for viral outbreaks and the development of microbial resistance. By comparing the antibacterial activity and determining the biosafety of ZnO-NPs compared to Sterillium, a commercial alcohol-based hand sanitizer, ZnO-NPs demonstrated their superior antibacterial activity and compatibility with the biological system as revealed by MIC values and TEM studies. Gram-positive bacteria were more susceptible to ZnO-NPs in comparison to Gram-negative bacteria. However, the highest MIC value of ZnO-NPs required to kill Gram-negative bacteria is demonstrated to be compatible with the biological system. Taken together, ZnO-NPs could be a promising candidate for hand sanitation in comparison to alcohol-based hand sanitizers; however, further studies related to the long-term toxicity and stability of ZnO-NPs, as well as investigations into their antimicrobial activity and safety in healthcare settings, are still required in the future to ascertain their antimicrobial activity and safety.

## Figures and Tables

**Figure 1 antibiotics-11-01606-f001:**
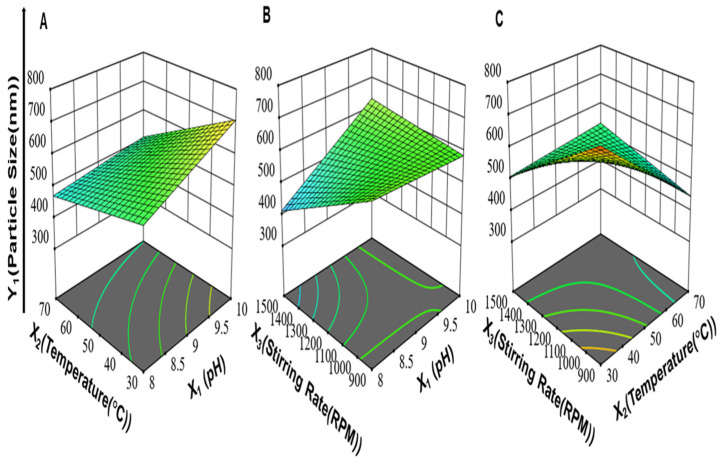
The 3-D response surface plots for (**A**) the interaction between pH and reaction temperature (X_1_X_2_), (**B**) the interaction between pH and stirring rate (X_1_X_3_), (**C**) and the interaction between reaction temperature and stirring rate (X_2_X_3_) on particle size (Y_1_) of ZnO-NPs while keeping the third variable at an average value between its high and low levels.

**Figure 2 antibiotics-11-01606-f002:**
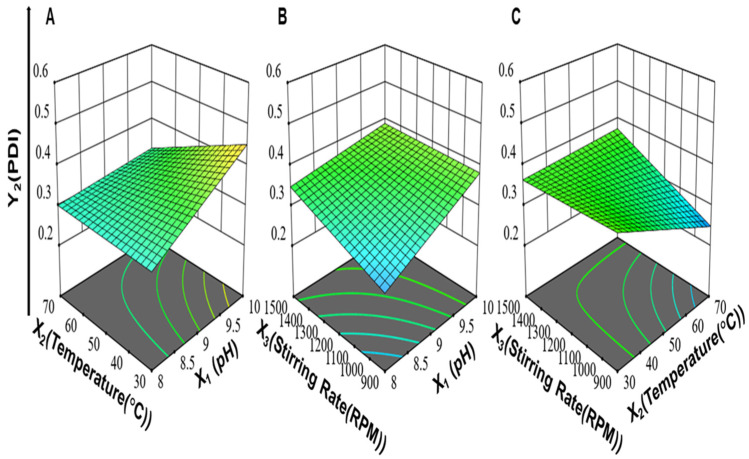
The 3-D response surface plots for (**A**) the interaction between pH and reaction temperature (X_1_X_2_), (**B**) the interaction between pH and stirring rate (X_1_X_3_), and (**C**) the interaction between reaction temperature and stirring rate (X_2_X_3_) on PDI (Y_2_) of ZnO NPs while keeping the third variable at an average value between its high and low levels.

**Figure 3 antibiotics-11-01606-f003:**
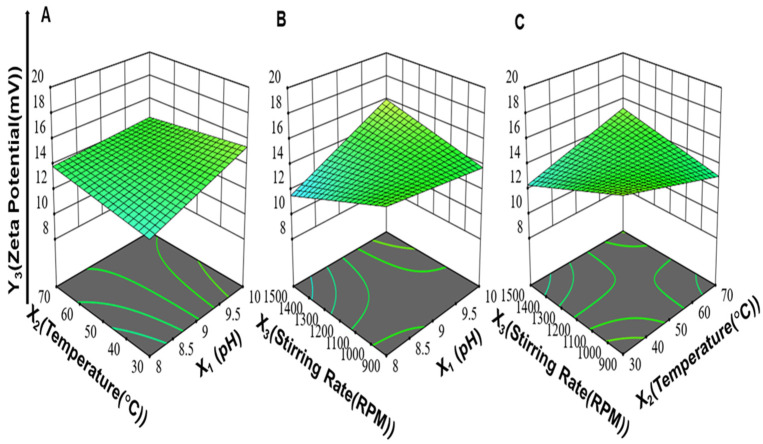
3-D response surface plots for (**A**) the interaction between pH and reaction temperature (X_1_X_2_), (**B**) the interaction between pH and stirring rate (X_1_X_3_), and (**C**) the interaction between reaction temperature and stirring rate (X_2_X_3_) on zeta potential of ZnO NPs while keeping the third variable at an average value between its high and low levels.

**Figure 4 antibiotics-11-01606-f004:**
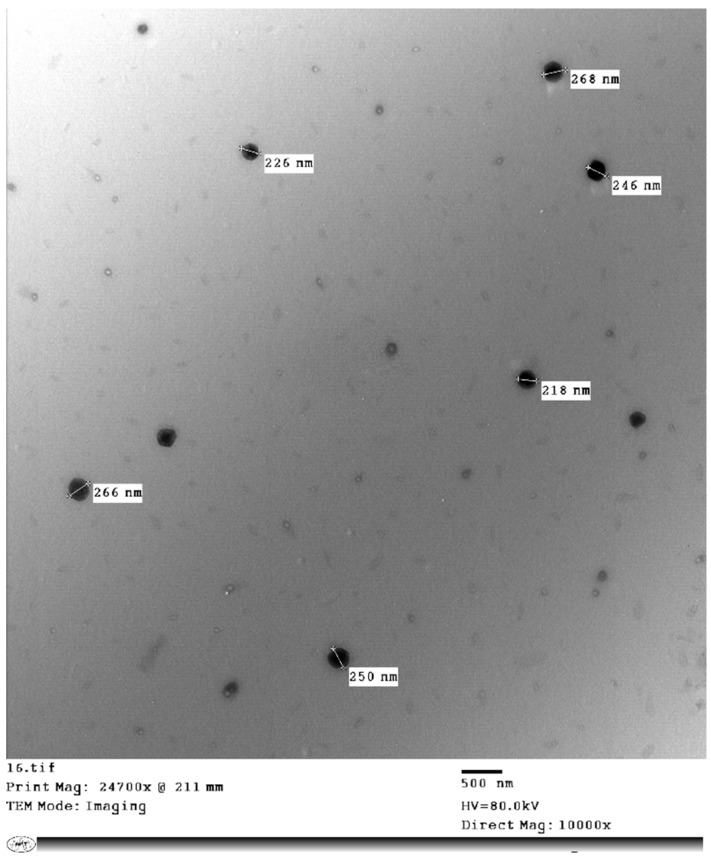
TEM images of synthesized ZnO-NPs.

**Figure 5 antibiotics-11-01606-f005:**
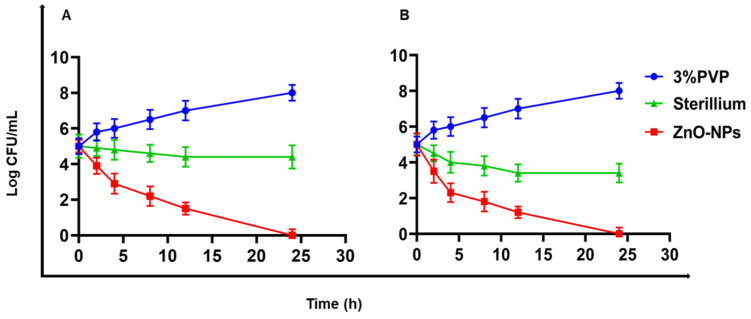
Time–kill curve of antibacterial activity of the ZnO-NPs and Sterillium at their MIC value against (**A**) MRSA and (**B**) *A. baumannii*. Each time point is the average of two independent experiments with three replicates each. Error bars represent standard deviation.

**Figure 6 antibiotics-11-01606-f006:**
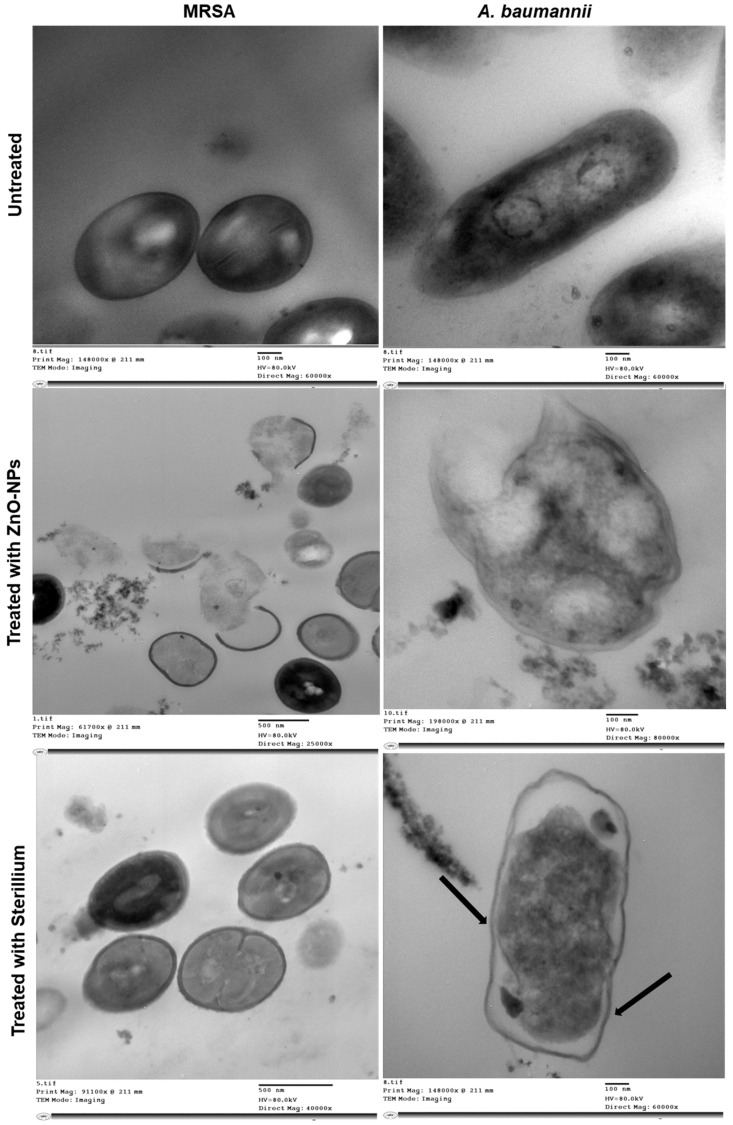
TEM images presenting the structural changes in MRSA and *A. baumannii* when treated with ZnO-NPs and Sterillium at a concentration less than MIC in comparison to untreated bacteria. As revealed, untreated bacteria showed a normal cell wall outline with no itching or degradation with the appearance of normal cytoplasm and intracellular components. However, bacterial damage was recorded in both bacteria treated with ZnO-NPs, and this involved damage to cell wall and release of intracellular contents, leading to bacterial death. In contrast, no structural changes were observed in MRSA after Sterillium treatment and this confirmed the lack of efficacy of Sterillium against MRSA; however, only morphological changes were observed with *A. baumannii,* such as appearance of body projection, as indicated by black arrow.

**Figure 7 antibiotics-11-01606-f007:**
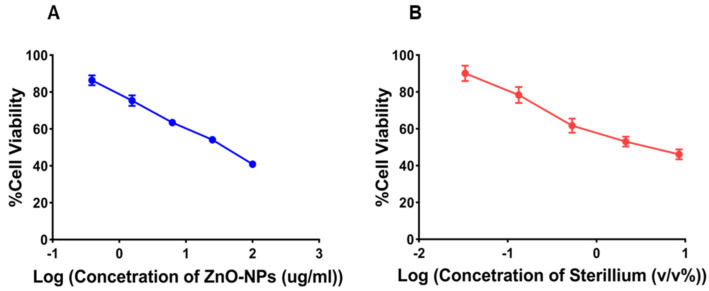
MTT cytotoxicity study of (**A**) ZnO-NPs and (**B**) a market product (Sterillium) on human dermal fibroblast cells. Each point is the average of two independent experiments, each performed in triplicate. Error bars represent standard deviation.

**Figure 8 antibiotics-11-01606-f008:**
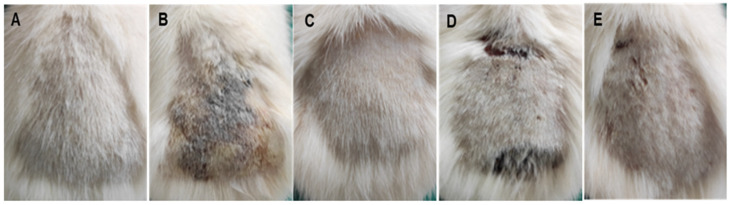
Representative photos of gross examination for representative rat skin of (**A**) Group I (negative control, shaved animal skin was not swapped with *S. aureus,* and without applying any tested samples), revealed normal skin; (**B**) Group II (positive control, animal skin was swapped only with *S. aureus,* and remained untreated), which revealed thickening, scaly dark-colored skin with a deep peeled ulcer covered by crust; (**C**) Group III (shaved animal skin was swapped with *S. aureus* followed by swapping with ZnO-NPs group), which revealed normal skin appearance without any signs of inflammation and was identical to negative control; (**D**) Group IV (shaved skin was swapped with *S. aureus* and then swapped with 3% PVP) showed scaly thick ulcerated skin similar to positive control; and (**E**) Group V (shaved animal skin was swapped with *S. aureus* and then swapped with Sterillium), which revealed mild erythema with mild scales formation.

**Figure 9 antibiotics-11-01606-f009:**
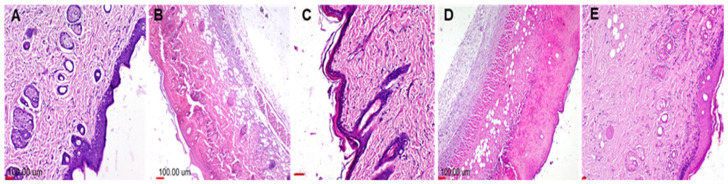
Representative photos of H&E-stained photomicrographs for representative rat skin of (**A**) Group I (negative control, shaved animal skin was not swapped with *S. aureus*, with no samples swapping), which showed normal skin tissue; (**B**) Group II (positive control, animal skin was swapped only with *S. aureus*, and remained untreated), which showed epidermal necrosis with pleomorphic inflammatory cell infiltration and dead follicular sheets; (**C**) Group III (shaved animal skin was swapped with *S. aureus* followed by swapping with ZnO-NPs), which showed normal skin structure; (**D**) Group IV (shaved skin was swapped with *S. aureus* and then swapped with 3% PVP), which showed severe necrotic crust in epidermal area with massive pleomorphic inflammatory cell infiltration and vacuolar degeneration of skin muscles; and (**E**) Group V (shaved animal skin was swapped with *S. aureus* and then swapped with Sterillium), which showed mild inflammatory cell infiltration on the epidermal–dermal junction with limited infiltration into the dermis layer. This is indicative of the superior antibacterial activity and biosafety of ZnO-NPs compared to Sterillium.

**Figure 10 antibiotics-11-01606-f010:**
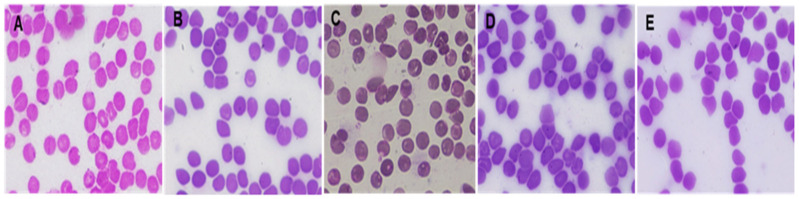
Representative photos of blood film for one representative rat blood film of (**A**) Group I (negative control, shaved animal skin was not swapped with *S. aureus,* with no samples swapping), (**B**) Group II (positive control, animal skin was swapped only with *S. aureus,* and remained untreated), (**C**) Group III (shaved animal skin was swapped with S. aureus followed by swapping with ZnO-NPs group), (**D**) Group IV (shaved skin was swapped with *S. aureus* and then swapped with 3% PVP), and (**E**) Group V, (shaved animal skin was swapped with bacteria and then after swapped with Sterillium), all groups revealed no abnormalities or alteration in shape of RBCs when compared to negative control and this is indicative of absence of any harmful effect for both ZnO-NPs and Sterillium.

**Figure 11 antibiotics-11-01606-f011:**
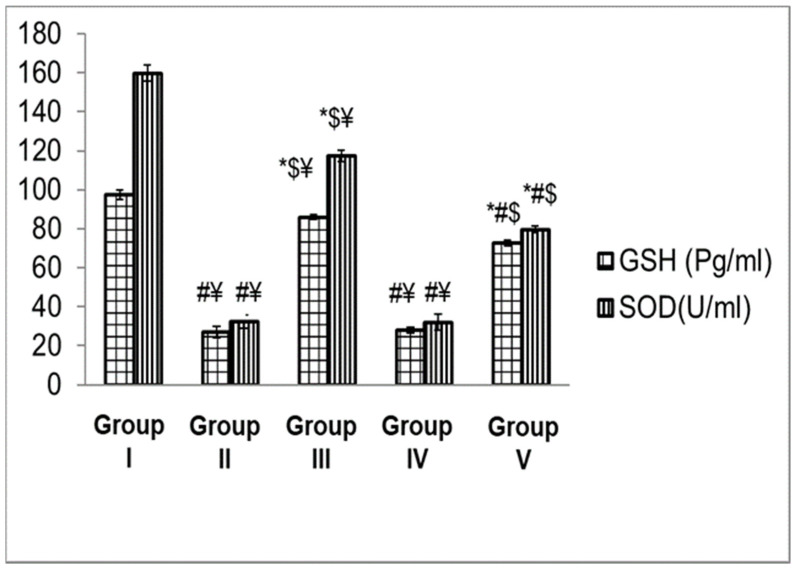
Levels of oxidative stress markers (GSH and SOD) determined for Group I (negative control, shaved animal skin was not swapped with *S. aureus,* and no samples were swapped afterwards), Group II (positive control, animal skin was swapped only with *S. aureus,* and remained untreated), Group III (shaved animal skin was swapped with S. aureus followed by swapping with ZnO-NPs group), Group IV (shaved skin was swapped with *S. aureus* and then swapped with 3% PVP), and Group V, (shaved animal skin was swapped with bacteria and then after swapped with Sterillium). Data are expressed as average of results, which are the average of five independent readings. Error bars represent standard error. Data were analyzed by one-way ANOVA. *, #, $, and ¥ represent significant differences at *p* < 0.05, where * represents significant differences when compared to positive control group, # represents significant differences when compared to ZnO-NPs group, $ represents significant differences when compared to 3% PVP group, ¥ represents significant differences when compared to Sterillium.

**Table 1 antibiotics-11-01606-t001:** Factors and levels that affected the particle size, polydispersity, and surface charge of ZnO-NPs.

Factors	Levels	
	Low (−1)	High (+1)
X_1_: pH	8	10
X_2_: Reaction temperature (°C)	30	70
X_3_: Stirring rate (RPM)	900	1500
**Responses**	**Units**	**Constraints**
Y_1_: Particle size	Nm	Minimum
Y_2_: PDI	-	Minimum
Y_3_: Zeta potential	mV	Maximum

**Table 2 antibiotics-11-01606-t002:** Experimental data values for particle size, polydispersity index, and zeta potential obtained by 2^3^ Full factorial design. Data are shown as mean ± standard deviation. Results are averages of two independent experiments with three replicates each.

Formula Code	pH	Reaction Temperature (°C)	Stirring Rate (RPM)	Particle Size (Dnm ± SD)	Polydispersity Index (PDI) ± SD	Zeta Potential (mV ± SD)
**F1**	8	30	900	741.63 ± 36.04	0.265 ± 0.02	−13.77 ± 0.35
**F2**	8	30	1500	379.60 ± 39.80	0.33 ± 0.03	−11.56 ± 1.53
**F3**	8	70	900	496.27 ± 20.11	0.23 ± 0.04	−16.20 ± 0.70
**F4**	8	70	1500	445.27 ± 13.46	0.37 ± 0.04	−11.57 ± 0.12
**F5**	10	30	900	775.80 ± 24.70	0.50 ± 0.01	−17.67 ± 0.76
**F6**	10	30	1500	638.47 ± 41.95	0.40 ± 0.02	−13.17 ± 0.40
**F7**	10	70	900	397.40 ± 1.71	0.26 ± 0.02	−9.94 ± 0.32
**F8**	10	70	1500	622.27 ± 13.48	0.38 ± 0.02	−18.87 ± 0.15

**Table 3 antibiotics-11-01606-t003:** Validation of the optimum synthesized ZnO-NPs (F3).

Response	Predicted Value	Observed Value	% Error
**Particle size (nm)**	489.871	496.27	1.31%
**PDI (nm)**	0.233	0.233	Zero%
**Zeta potential (mV)**	16.2	16.2	Zero%

**Table 4 antibiotics-11-01606-t004:** * MIC values of ZnO-NPs (µg/mL) and Sterillium (*v/v*%) recorded for *S. aureus,* MRSA, *K. pneumoniae*, and *A. baumannii,* and their ** CC50 toward human dermal fibroblast cells (HDFa).

Formulations	MIC			
	*S. aureus*	*MRSA*	*K. pneumoniae*	*A. baumannii*
**ZnO-NPs**	180	350	455	625
**Sterillium**	85	*** NE	68	59.5

* MIC: Minimum inhibitory concentration; ** CC50: Cytotoxic concentrations of ZnO-NPs and Sterillium responsible for the death of 50% of HDFa is equivalent to 35.32 ± 1.86 (µg/mL) and 3.686 ± 0.195 (*v/v*%), respectively. *** NE: Not effective.

**Table 5 antibiotics-11-01606-t005:** Hirasawa scoring of histopathological alterations in skin.

	Groups	NegativeControl	Positive Control	ZnO-NPs	PVP (3% *w*/*v*)	Sterillium
**Epidermis**	Hypertrophy	0	0	0	0	0
	Hyperkeratosis	0	0	0	0	0
	Parakeratosis	0	0	0	0	0
	Erosion	0	3 ^b,d^	0 ^a,c,d^	3 ^b,d^	1 ^a,b,c^
	Inflammatory cells infiltration	0	3 ^b,d^	0 ^a,c,d^	3 ^b,d^	1 ^a,b,c^
	Extracellular edema	0	0	0	0	0
**Corium**	Ulcer	0	3 ^b,d^	0 ^a,c,d^	3 ^b,d^	0 ^a,b,c^
	Inflammatory cells infiltration	0	3 ^b,d^	0 ^a,c,d^	3 ^b,d^	1 ^a,b,c^
**Subcutis**	Inflammatory cells infiltration	0	3 ^b,d^	0 ^a,c,d^	2 ^b,d^	0 ^a,b,c^

Hirasawa scoring of histopathological alterations in skin; data are presented as median using Kruskal–Wallis test followed by the Mann–Whitney U test. Data were analyzed by using one-way ANOVA. ^a–d^ were significantly different at *p* < 0.05, where ^a^ significantly differs when compared to positive control group, ^b^ significantly differs when compared to ZnO-NPs group, ^c^ significantly differs when compared to group treated with PVP (3% *w*/*v*), and ^d^ significantly differs when compared to market product (Sterillium).

**Table 6 antibiotics-11-01606-t006:** Erythrogram of different experimental groups; * data are presented as mean ± Standard Error (SE) and results are averages of five independent readings.

Animal Groups	PCV (%) ± SE	Hb(g/dL) ± SE	RBCs (×10^6^/nL) ± SE	MCV (fL) ± SE	MCHC (%) ± SE
**Negative Control**	39.53 ± 0.246	12.06 ± 0.635	7.84 ± 0.107	48.87 ± 0.483	37.58 ± 0.409
**Positive Control**	39.41 ± 0.247	12.34 ± 0.199	7.61 ± 0.140	48.72 ± 0.645	37.64 ± 0.354
**ZnO-NPs**	38.96 ± 0.325	12.60 ± 0.134	7.99 ± 0.112	47.92 ± 0.299	37.8 ± 0.224
**3% PVP**	39.53 ± 0.246	13.07 ± 0.418	7.08 ± 0.270	48.51 ± 0.549	38.08 ± 0.445
**Sterillium**	39.96 ± 0.233	13.42 ± 0.306	8.34 ± 0.186	48.22 ± 0.481	38.06 ± 0.209

* Data were analyzed by One-way ANOVA and there was a non-significant difference between untreated (control) and treated animals at *p* > 0.05. PCV, packed cell volume, Hb = hemoglobin (g/dL = gram/deciliter), RBCs = red blood cells count, MCV = main corpuscular volume (fL, femto-liter), MCHC = main corpuscular hemoglobin concentrations.

## Data Availability

All authors are happy to share all data (including Supplementary Data).

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
