# Peer review of "Investigating the Antibacterial Activity and Safety of Zinc Oxide Nanoparticles versus a Commercial Alcohol-Based Hand-Sanitizer: Can Zinc Oxide Nanoparticles Be Useful for Hand Sanitation?"

_antibiotics, 2022, doi:10.3390/antibiotics11111606_

Round 1
Reviewer 1 Report
Authors have presented a high value scientific research with novel hypotheses and outcomes. The objective of the research is well supported with various experiments and has significant impact. Overall, authors have synthesized a ZnO NP formulation and presented in-vitro and in-vivo studies. They have concluded the superior anti-bacterial efficacy of ZnO NP against marketed formulation containing alcohol using several microorganisms. MTT and in-vivo results are well presented and discussed in depth. Authors have shown great efforts in the current work. However, there are few general comments and suggestions to improve the final quality of the manuscript.
1. In the synthesis of ZnO-NP, section 2.2.1, why did authors follow oven based method for drying NP and not lyophilization/ freeze drying? The formulation contains PEG-600, what about stability of the same during drying at high temperature?
2. Authors have used 3% PVPK25 solution for dispersion. Why only 3% in particular? Did you determine sedimentation/stability of ZnO-NP in this solution?
3. Section 2.25.2 and section 3.4, use clear and small sentence to make the grouping between animals logical. Treatment to different groups is very confusing. Reframe the sentences.
4. Section 3.1, DSC results, authors have stated that endothermic peak between 200- 350 is due to release of water molecules. In majority of DSC findings, such event occur between 80-120 C. Can you support your statement with more references?
5. In FTIR results, authors need to discuss more on the events taking place in the finger-print region
6. In table 3, is the Zeta value is -ve or +ve?
Author Response
Reviewer comments
Reviewer 1
Open Review
( ) I would not like to sign my review report
(x) I would like to sign my review report
English language and style
( ) Extensive editing of English language and style required
(x) Moderate English changes required
( ) English language and style are fine/minor spell check required
( ) I don't feel qualified to judge about the English language and style
Yes |
Can be improved |
Must be improved |
Not applicable |
|
Does the introduction provide sufficient background and include all relevant references? |
(x) |
( ) |
( ) |
( ) |
Are all the cited references relevant to the research? |
(x) |
( ) |
( ) |
( ) |
Is the research design appropriate? |
(x) |
( ) |
( ) |
( ) |
Are the methods adequately described? |
( ) |
(x) |
( ) |
( ) |
Are the results clearly presented? |
(x) |
( ) |
( ) |
( ) |
Are the conclusions supported by the results? |
(x) |
( ) |
( ) |
( ) |
Comments and Suggestions for Authors
Authors have presented a high value scientific research with novel hypotheses and outcomes. The objective of the research is well supported with various experiments and has significant impact. Overall, authors have synthesized a ZnO NP formulation and presented in-vitro and in-vivo studies. They have concluded the superior anti-bacterial efficacy of ZnO NP against marketed formulation containing alcohol using several microorganisms. MTT and in-vivo results are well presented and discussed in depth. Authors have shown great efforts in the current work. However, there are few general comments and suggestions to improve the final quality of the manuscript.
Please improve the following remarks:
Reviewer1, Comment 1:
1.In the synthesis of ZnO-NP, section 2.2.1, why did authors follow oven-based method for drying NP and not lyophilization/ freeze drying? The formulation contains PEG-600, what about stability of the same during drying at high temperature?
Response to reviewer 1, Comment 1
Thanks for reviewer comments, the PEG stability was not affected by drying the sample at oven (150°C) , the boiling point for PEG 600 is >200°C kindly check safety data sheet for PEG 600 provided by sigma (https://www.sigmaaldrich.com/EG/en/sds/sial/81180). Moreover, the peaks characterizing PEG600 adsorbed at the surface of ZnO-NPs were identified in Figure S2 and Figure S3 in supplementary file for DSC and FTIR respectively. Therefore, using oven drying is considered as a quick method for drying and doesn’t have any harmful effect on components forming ZnO-NPs.
Reviewer1, Comment 2:
- Authors have used 3% PVPK25 solution for dispersion. Why only 3% in particular? Did you determine sedimentation/stability of ZnO-NP in this solution?
Response to reviewer 1, Comment 2
Thanks for reviewer comment, basically, two concentrations of PVP25K were investigated to assure stability of ZnO-NPs after tracking the particle size for several days after preparing ZnO- NPs dispersion and the data for particle size, polydispersity and zeta potential are provided in supplementary file (Table S1) and the table is copied below for reviewer convenience. Additionally, we have shed the light on this in the manuscript, please check page 9 line 392 to 394.
Table S1: *Stability of ZnO-NPs prepared under different PVP25K concentrations, results are average of three independent experiments, each involved a triplicate |
|||
PVP concentration- Day of measurement after preparation |
Particle Size (Dnm ± SD) |
Polydispersity Index (PDI) ± SD |
Zeta Potential (mV ± SD) |
3% w/v – Day 0
|
361.7±2.08 |
0.302±0.008 |
-18.2±0.14 |
3% w/v – Day 1
|
378.7±2.25 |
0.323±0.03 |
-14.93±0.5 |
3% w/v – Day 2
|
395.3±13.5 |
0.33±0.043 |
-11.6±0.11 |
3% w/v – Day 3
|
487.4±46 |
0.34±0.019 |
-10.4±0.25 |
10% w/v - Day 0
|
389.6±11 |
0.311±0.006 |
-10.6±0.2 |
10% w/v - Day 1
|
458±14.3 |
0.345±0.045 |
-13.17±0.153 |
10% w/v - Day 2
|
503±6 |
0.29±0.025 |
-13.6±0.47 |
10% w/v - Day 3
|
501.6±4.56 |
0.288±0.025 |
-13.63±0.47 |
*Preparation Condition; pH=8, Temperature = 70°C, RPM = 1500 |
Reviewer1, Comment 3:
- Section 2.25.2 and section 3.4, use clear and small sentence to make the grouping between animals logical. Treatment to different groups is very confusing. Reframe the sentences.
Response to reviewer 1, Comment 3
Thanks for reviewer comment but we believe that giving a detailed expression of each group is essential to help the reader to follow the data presented in animal study.
Reviewer1, Comment 4:
- Section 3.1, DSC results, authors have stated that endothermic peak between 200- 350 is due to release of water molecules. In majority of DSC findings, such event occur between 80-120 C. Can you support your statement with more references?
Response to reviewer 1, Comment 4
Thanks for reviewer comment, we have doubled checked the data and we have updated this section is copied below for reviewer convenience; kindly check page 8, line 361 to 369.
ZnO-NPs thermal stability was presented in Figure S2 in supplementary file, ZnO-NPs showed high thermal stability in terms of the overall weight loss obtained at the end of the testing time and temperatures up to 1000 °C. In particular, ZnO-NPs showed only 0.2187wt% weight loss at temperature up to 200 °C, and this might be explained by loss of water molecules that are bound to the surface of ZnO-NPs and this is consistent with literature (Spoială et al. 2021). However, the total weight loss at the end of test was 2.459 wt% and this might be due to release of water molecules that are chemically combined with PEG, in addition to decomposition of PEG 600 molecules adsorbed at the surface of ZnO-NPs. This is matched with what was reported in literature (Dilshad et al. 2017).
Reviewer1, Comment 5:
- In FTIR results, authors need to discuss more on the events taking place in the finger-print region
Response to reviewer 1, Comment 5
Thanks for reviewer comment, we have updated this section with more peaks and highlighted in yellow, and this section is copied below for reviewer convenience, kindly check page 9, line 377 to line 381
FTIR spectra of ZnO-NPs was presented in Figure S3 in supplementary file, bands at about 500 and 435.8 cm-1 are attributed to the formation of the stretching vibration of metal-oxygen (Zn–O) bonds. This was matched with literature where the infra-red characteristic bands of ZnO-NPs were reported to be in the region from 680 up to 300 cm−1 (Kajbafvala et al. 2010). The band recorded at 3421.7 cm−1 was characteristic for the stretching vibration of intermolecular hydrogen bond (OH) existing between the adsorbed water molecules (Winiarski et al. 2018). In addition, this band could be attributed to PEG molecules adsorbed onto the surface of ZnO-NPs (Xu et al. 2014; Varughese et al. 2020). While bands at 1617.8 and 1033 cm-1 could be correlated to stretching vibration of (-C-O-C-) ether groups and (COO-) carboxylate groups, respectively for PEG molecules (Xu et al. 2014; Tai et al. 2016).
Reviewer1, Comment 6
- In table 3, is the Zeta value is -ve or +ve?
Response to reviewer 1, Comment 6
The zeta potential values are provided as negative values.

Reviewer 2 Report
The manuscript describes the synthesis, characterization of Zinc Oxide nanoparticles, and more importantly, their antibacterial activity and safety as for hand sanitation. It is an interesting topic which I consider of interest for readers of Antibiotics.
The manuscript needs minor revision before its accepted for publication.
Comments:
- page 20, and Table 4,
CC50 value for Sterillium is 3.686? while for Zn-NPs is 35.32.
- Fig. 1 UV, Fig. 2 DSC-TGA, Fig. 3 FTIR, Fig. 4 X-ray diffraction can be moved to the supplementary materials, as these figs are the characterizations of ZnO nps, no need to put them in the main text.
Author Response
Reviewer2
Open Review
( ) I would not like to sign my review report
(x) I would like to sign my review report
English language and style
( ) Extensive editing of English language and style required
( ) Moderate English changes required
(x) English language and style are fine/minor spell check required
( ) I don't feel qualified to judge about the English language and style
Yes |
Can be improved |
Must be improved |
Not applicable |
|
Does the introduction provide sufficient background and include all relevant references? |
(x) |
( ) |
( ) |
( ) |
Are all the cited references relevant to the research? |
(x) |
( ) |
( ) |
( ) |
Is the research design appropriate? |
(x) |
( ) |
( ) |
( ) |
Are the methods adequately described? |
(x) |
( ) |
( ) |
( ) |
Are the results clearly presented? |
(x) |
( ) |
( ) |
( ) |
Are the conclusions supported by the results? |
(x) |
( ) |
( ) |
( ) |
Comments and Suggestions for Authors
The manuscript describes the synthesis, characterization of Zinc Oxide nanoparticles, and more importantly, their antibacterial activity and safety as for hand sanitation. It is an interesting topic which I consider of interest for readers of Antibiotics.
The manuscript needs minor revision before its accepted for publication.
Comments
Reviewer 2, Comment 1:
- page 20, and Table 4,
CC50 value for Sterillium is 3.686? while for Zn-NPs is 35.32.
Response to reviewer 2, Comment 1
The question of the reviewer is not clear for authors, however, as we have stated in our response to reviewer4, comment 6;
We would like to confirm that we cannot directly compare between Sterillium and ZnO-NPs as MIC values for each one is presented in different unit where it is expressed as v/v for Sterillium and as µg/ml for ZnO-NPs. However, the comparison between each them was based on the outcome of antibacterial activity, their cytotoxicity as well as animal study. As discussed in the manuscript, sterillium (original solution available in the market) cannot eradicate MRSA, however, ZnO-NPs can completely eradicate MRSA. In addition, MIC for tested bacteria in comparison to CC50 of each product was assessed and as presented and discussed along the body text of manuscript; ZnO-NPs showed a better antibacterial activity (eradication of MRSA) and was safer. Moreover, animal study revealed a complete eradication of bacteria with the absence of any histopathological alterations in case of ZnO-NPs contrary to histopathological changes identified in Sterillium. Kindly check the highlighted section in Page 14, line 538 to 547 and page 17 and 18, line 636 to line 658.
Reviewer 2, Comment 2:
- Fig. 1 UV, Fig. 2 DSC-TGA, Fig. 3 FTIR, Fig. 4 X-ray diffraction can be moved to the supplementary materials, as these figs are the characterizations of ZnO nps, no need to put them in the main text.
Response to reviewer 2, Comment 2
Thanks for reviewer comments, all Figures concerning ZnO-NPs characterization were moved to supplementary file and the Figures numbers are updated in the body of the manuscript.

Reviewer 3 Report
The authors have proposed an interesting topic regarding the antibacterial effect of ZnO-NPs as a novel hand sanitizer materials. The paper indicated the antimicrobial effect from the view of Materials Science, which is integrity. However, some concepts regarding the dose assessment and exposure testing for the animal are not included in this paper, which reduces the scientific soundness of the article.
ZnO is believed to be toxic and the ZnO NPs are considered to be able to accumulate in the body and indicate long-term toxicity. The current exposure design is too short for such an effect to be indicated. Also you ZnO NPs might be highly concentrated for the living organs as well.
Suggestion for the improvement to be purely considered the antibacterial effect rather than perform the comparison with the hand santilizer or limit the application into this aspect. You can refer to one article for the reasonable application with limited effect of toxins. e.g., Rapid preparation and antimicrobial activity of polyurea coatings with RE-Doped nano-ZnO https://doi.org/10.1111/1751-7915.13891
Last but not the least, the design of the experiment can also be concluded into 2 segments, one is regarding bacteria level, the other is a field test, then you can do such compilation to clear up your methodology chapter.
Author Response
Reviewer 3
Open Review
(x) I would not like to sign my review report
( ) I would like to sign my review report
English language and style
( ) Extensive editing of English language and style required
(x) Moderate English changes required
( ) English language and style are fine/minor spell check required
( ) I don't feel qualified to judge about the English language and style
Yes |
Can be improved |
Must be improved |
Not applicable |
|
Does the introduction provide sufficient background and include all relevant references? |
( ) |
(x) |
( ) |
( ) |
Are all the cited references relevant to the research? |
( ) |
(x) |
( ) |
( ) |
Is the research design appropriate? |
( ) |
( ) |
(x) |
( ) |
Are the methods adequately described? |
( ) |
(x) |
( ) |
( ) |
Are the results clearly presented? |
( ) |
( ) |
(x) |
( ) |
Are the conclusions supported by the results? |
( ) |
(x) |
( ) |
( ) |
Comments and Suggestions for Authors
Reviewer 3, Comment 1:
The authors have proposed an interesting topic regarding the antibacterial effect of ZnO-NPs as a novel hand sanitizer materials. The paper indicated the antimicrobial effect from the view of Materials Science, which is integrity. However, some concepts regarding the dose assessment and exposure testing for the animal are not included in this paper, which reduces the scientific soundness of the article.
Response to reviewer 3, Comment 1
Thanks for reviewer comment; regarding the dose, we have determined MIC values for each micro-organism as provided in Table 4 – these concentrations are suggested to be used for hand sanitation. Moreover, CC50 was determined for ZnO-NPs and Sterillium, that is approved to be used in hospitals for hand sanitation and as revealed from data provided in page 17 and 18, that ZnO-NPs had an effective antibacterial activity and they are safer than Sterillium. Moreover, we run short-term animal study and as revealed from histopathology data (Page19, Figure 9), there is no histopathological alterations observed in ZnO-NPs treated group compared to untreated group. This could be considered as a promising data regarding the safety and antibacterial activity of ZnO-NPs. However, we plan to provide a serial of publications concerning the stability of ZnO-NPs, chronic toxicity in animal model after topical exposure, as well as, formulating ZnO-NPs in a gel a form as we believe that gel formulation will be more stable when compared to colloidal dispersion of ZnO-NPs and finally run a pilot study comparing the hand sanitation potential of ZnO-NPs compared to sterillium.
Reviewer 3, Comment 2:
ZnO is believed to be toxic and the ZnO NPs are considered to be able to accumulate in the body and indicate long-term toxicity. The current exposure design is too short for such an effect to be indicated. Also your ZnO NPs might be highly concentrated for the living organs as well.
Response to reviewer 3, Comment 2
The toxicity of ZnO-NPs is dependent on many factors including particle size, concentration used, exposure time, etc. In addition, several publications reported the safety of ZnO-NPs for topical application, kindly check the following references (Ryu et al. 2014; Mohammed et al. 2019)
As we mentioned in our response to reviewer 3, comment 2, we plan to do serial of publications to assure the safe application of ZnO-NPs for hand sanitation. As all investigations highlighted in our response to reviewer 3 comment 2, cannot be published in one article but it needs to be published in a serial of publications. We have a previous experience with this regarding our previous publications concerning rhamnolipids nano-micelles preparation and application for hand sanitation. Kindly check the following links for these publications;
Article 1: Rhamnolipids Nano-Micelles as a Potential Hand Sanitizer; https://www.mdpi.com/2079-6382/10/7/751
Article 2: Rhamnolipid Nano-Micelles versus Alcohol-Based Hand Sanitizer: A Comparative Study for Antibacterial Activity against Hospital-Acquired Infections and Toxicity Concerns; https://www.mdpi.com/2079-6382/11/5/605
Article 3: includes the antiviral activity of rhamnolipids nano-micelles against SARS-CoV2 and Acute study confirming its safety upon skin and eye exposure – this manuscript has been accepted in 30th October 2022 for publication.
Reviewer 3, Comment 3:
Suggestion for the improvement to be purely considered the antibacterial effect rather than perform the comparison with the hand santilizer or limit the application into this aspect. You can refer to one article for the reasonable application with limited effect of toxins. e.g., Rapid preparation and antimicrobial activity of polyurea coatings with RE-Doped nano-ZnO https://doi.org/10.1111/1751-7915.13891
Response to reviewer 3, Comment 3
Thanks for reviewer comment, but we would like to continue in our investigations and address the safe application of ZnO-NPs as a hand sanitizer. Then after, we can extend use of ZnO-NPs to other applications
Reviewer 3, Comment 4;
Last but not the least, the design of the experiment can also be concluded into 2 segments, one is regarding bacteria level, the other is a field test, then you can do such compilation to clear up your methodology chapter.
Response to reviewer 3, Comment 3
Thanks to reviewer comment, as we clarified in our response in the first comment, we plan to do this. In addition, we have edited the writing in the manuscript, kindly check Abstract line 33 to 36; Therefore, ZnO-NPs could be a promising candidate for hand sanitation in comparison to alcohol-based hand sanitizers; however, several studies related to long-term toxicity, stability of ZnO-NPs as well as investigating the antimicrobial activity and safety in healthcare settings are still required to be performed in the future to ascertain their antimicrobial activity and safety. In addition, kindly check the edited conclusion section in page 23 line 811 to 815

Reviewer 4 Report
Dear Authors,
The theme of your research is of interest for the discovery of a new active hand sanitizer, not alcohol based, that will eliminate the discomfort and skin problems related to the frequent use of alcohol-based hand sanitizers.
I have some corrections and also suggestions for your manuscript:
1. Reduce the Abstract. In the actual form, it is more an Introduction than abstract
2. There are some English grammar/language or spelling mistakes, for example in line 23 "are known", line 85 "production", line 90 "activity", line 109 "performed". "And" in lines 222, 240, 251, etc. should not be written in italic. Rephrase lines 108-110, 118-119, 648
3. Reduce the dimension of figures
4. For the methods' protocols, that are well-known, standardized, you should put a reference for the method, in order to avoid the overload of the text. Example: protocol for the MTT assay, lines 295-304.
5. You should put in italic "in vivo', "in vitro", "A. baumannii".
6. In the antibacterial test, the MICs registered for ZnO-NPs were higher than those for Sterilium, therefore the antibacterial activity of the nanoparticles is lower, not higher.
7. Redo Table 5 because there are data that are not visible
8. You do not discuss some extremely important aspects concerning the use of ZnO-NPs on skin: the pharmaceutical form, the formulation (colloidal suspension?), the stability of the nanoparticles suspension, the frequence of use, the risks of overuse, if there are restrictions for the use (different patients, skin problems, etc.), the interactions with different other products on skin, etc.
Strongly, I do not consider that you can conclude that this formulation may be used as hand sanitizer without some more investigations, some of them marked by me in point 8 of these comments.
Author Response
Reviewer 4
Open Review
(x) I would not like to sign my review report
( ) I would like to sign my review report
English language and style
( ) Extensive editing of English language and style required
( ) Moderate English changes required
(x) English language and style are fine/minor spell check required
( ) I don't feel qualified to judge about the English language and style
Yes |
Can be improved |
Must be improved |
Not applicable |
|
Does the introduction provide sufficient background and include all relevant references? |
( ) |
(x) |
( ) |
( ) |
Are all the cited references relevant to the research? |
( ) |
(x) |
( ) |
( ) |
Is the research design appropriate? |
( ) |
(x) |
( ) |
( ) |
Are the methods adequately described? |
( ) |
(x) |
( ) |
( ) |
Are the results clearly presented? |
( ) |
(x) |
( ) |
( ) |
Are the conclusions supported by the results? |
( ) |
(x) |
( ) |
( ) |
Comments and Suggestions for Authors
Dear Authors,
The theme of your research is of interest for the discovery of a new active hand sanitizer, not alcohol based, that will eliminate the discomfort and skin problems related to the frequent use of alcohol-based hand sanitizers.
I have some corrections and also suggestions for your manuscript:
Reviewer 4, Comment 1:
- Reduce the Abstract. In the actual form, it is more an Introduction than abstract
Response to reviewer 4, Comment 1
The abstract has been reduced
Reviewer 4, Comment 2:
- There are some English grammar/language or spelling mistakes, for example in line 23 "are known", line 85 "production", line 90 "activity", line 109 "performed". "And" in lines 222, 240, 251, etc. should not be written in italic. Rephrase lines 108-110, 118-119, 648(???)
Response to reviewer 4, Comment 2
Thanks for reviewer comment, all are corrected and highlighted in yellow.
Reviewer 4, Comment 3:
- Reduce the dimension of figures
Response to reviewer 4, Comment 3
Figures dimension were reduced
Reviewer 4, Comment 4:
- For the methods' protocols, that are well-known, standardized, you should put a reference for the method, in order to avoid the overload of the text. Example: protocol for the MTT assay, lines 295-304.
Response to reviewer 4, Comment 4
Thanks for reviewer comment, this section has been rewritten according to reviewer instructions, kindly check the highlighted section in yellow in page 7, line 282 – 299.
Reviewer 4, Comment 5:
- You should put in italic "in vivo', "in vitro", "A. baumannii".
Response to reviewer 4, Comment 5
All are placed in italic and highlighted in yellow
Reviewer 4, Comment 6:
- In the antibacterial test, the MICs registered for ZnO-NPs were higher than those for Sterilium, therefore the antibacterial activity of the nanoparticles is lower, not higher.
Response to reviewer 4, Comment 6
Thanks for reviewer comment, we would like to confirm that we cannot directly compare between Sterillium and ZnO-NPs as MIC values for each one is presented in different unit where it is expressed as v/v for Sterillium and as µg/ml for ZnO-NPs. However, the comparison between each them was based on the outcome of antibacterial activity, their cytotoxicity as well as animal study. As discussed in the manuscript, sterillium (original solution available in the market) cannot eradicate MRSA, however, ZnO-NPs can completely eradicate MRSA. In addition, MIC for tested bacteria in comparison to CC50 of each product was assessed and as presented and discussed along the body text of manuscript; ZnO-NPs showed a better antibacterial activity (eradication of MRSA) and was safer. Moreover, animal study revealed a complete eradication of bacteria with the absence of any histopathological alterations in case of ZnO-NPs contrary to histopathological changes identified in Sterillium. Kindly check the highlighted section in Page 14, line 538 to 547 and page 17 and 18, line 636 to line 658.
Reviewer 4, Comment 7:
- Redo Table 5 because there are data that are not visible
Response to reviewer 4, Comment 7
We have used letters instead to be readable
Reviewer 4, Comment 8:
- You do not discuss some extremely important aspects concerning the use of ZnO-NPs on skin: the pharmaceutical form, the formulation (colloidal suspension?), the stability of the nanoparticles suspension, the frequence of use, the risks of overuse, if there are restrictions for the use (different patients, skin problems, etc.), the interactions with different other products on skin, etc.
Strongly, I do not consider that you can conclude that this formulation may be used as hand sanitizer without some more investigations, some of them marked by me in point 8 of these comments.
Response to reviewer 4, Comment 8
Kindly check our response to reviewer 4 comment 1; we copied it here for reviewer convenience
Thanks for reviewer comment; regarding the dose, we have determined MIC values for each micro-organism as provided in Table 4 – these concentrations are suggested to be used for hand sanitation. Moreover, CC50 was determined for ZnO-NPs and Sterillium, that is approved to be used in hospitals for hand sanitation and as revealed from data provided in page 17 and 18, that ZnO-NPs had an effective antibacterial activity and they are safer than Sterillium. Moreover, we run short-term animal study and as revealed from histopathology data (Page 19, Figure9), there is no histopathological alterations observed in ZnO-NPs treated group compared to untreated group. This could be considered as a promising data regarding the safety and antibacterial activity of ZnO-NPs. However, we plan to provide a serial of publications concerning the stability of ZnO-NPs, chronic toxicity in animal model after topical exposure, as well as, formulating ZnO-NPs in a gel a form as we believe that gel formulation will be more stable when compared to colloidal dispersion of ZnO-NPs and finally run a pilot study comparing the hand sanitation potential of ZnO-NPs compared to sterillium

Round 2
Reviewer 3 Report
The content can be accepted in current form.
But please re-align all the figures and crop the legend from the machine and use your own legend instead. The fonts are suggested to use the same one with the same size (recommended Arial).
Author Response
Comments and Suggestions for Authors
The content can be accepted in current form.
But please re-align all the figures and crop the legend from the machine and use your own legend instead. The fonts are suggested to use the same one with the same size (recommended Arial).
Response to reviewer 3
Thanks for reviewer comments, all figures are corrected following reviewer recommendation.
Reviewer 4 Report
Dear Authors,
Thank you for considering my suggestions and corrections. Still, I consider that you should add more data regarding the formulation of the gel, the dose used, the frequency of applications and more pharmacokinetics.
Author Response
Comments and Suggestions for Authors
Thank you for considering my suggestions and corrections. Still, I consider that you should add more data regarding the formulation of the gel, the dose used, the frequency of applications and more pharmacokinetics.
Response to reviewer 4
Thanks for reviewer comment. However, formulating ZnO-NPs in a gel form is out of the scope of this manuscript but it is rather considered as another piece of work that could be performed in the future by our team.
We do appreciate reviewers’ comments and hope that we managed to answer each point clearly
Thank you for your consideration of this manuscript.